# Tyrosinase Inhibitors Naturally Present in Plants and Synthetic Modifications of These Natural Products as Anti-Melanogenic Agents: A Review

**DOI:** 10.3390/molecules28010378

**Published:** 2023-01-02

**Authors:** Mubashir Hassan, Saba Shahzadi, Andrzej Kloczkowski

**Affiliations:** 1The Steve and Cindy Rasmussen Institute for Genomic Medicine, Nationwide Children’s Hospital, Columbus, OH 43205, USA; 2Department of Pediatrics, The Ohio State University College of Medicine, Columbus, OH 43205, USA

**Keywords:** melanogenesis, melanin, synthetic inhibitors, chemical compounds, anti-tyrosinase agents

## Abstract

Tyrosinase is a key enzyme target to design new chemical ligands against melanogenesis. In the current review, different chemical derivatives are explored which have been used as anti-melanogenic compounds. These are different chemical compounds naturally present in plants and semi-synthetic and synthetic compounds inspired by these natural products, such as kojic acid produced by several species of fungi; arbutin—a glycosylated hydroquinone extracted from the bearberry plant; vanillin—a phenolic aldehyde extracted from the vanilla bean, etc. After enzyme inhibition screening, various chemical compounds showed different therapeutic effects as tyrosinase inhibitors with different values of the inhibition constant and IC_50_. We show how appropriately designed scaffolds inspired by the structures of natural compounds are used to develop novel synthetic inhibitors. We review the results of numerous studies, which could lead to the development of effective anti-tyrosinase agents with increased efficiency and safety in the near future, with many applications in the food, pharmaceutical and cosmetics industries.

## 1. Introduction

Melanin is a darkish macromolecular pigment present in bacteria, fungi, insects, plants, invertebrates, and vertebrates [1,2,3]. In mammals, there are sorts of melanin, namely, eumelanin and pheomelanin, that are accountable for brown-to-black and yellow-to-pink colorations, respectively [4,5,6]. The basic characteristic of melanin in animals is the representation of apparent colors in the skin, hair, feathers, and pupils [7]. Melanin is secreted with the aid of using melanocytes positioned within side the basal epidermal layer [8]. Melanosomes are organelles discovered in melanocytes that biosynthesize melanin by using of a method called melanogenesis, which entails a series of complicated enzymatic and chemical processes [9]. The capabilities of melanocytes are managed by using intrinsic factors, including α-melanocyte stimulating hormone, and extrinsic factors, including chemical compounds and UV rays. Although melanin protects pores and skin from UV radiation and from different chemical compounds, its overaccumulation can cause hyperpigmentation-related diseases, esthetic problems and even skin cancer [10].

Tyrosinase is a copper-containing enzyme successfully used as an inhibitor for the treatment of melanogenesis [11]. Mushroom tyrosinase is a soluble tetrameric enzyme discovered within the cytoplasm, while mammalian tyrosinases, including human tyrosinase, are glycosylated monomeric enzymes anchored to the melanosome membrane [12]. Comparative sequence analysis showed that mushroom tyrosinase has 22–24% sequence identity with mammalian tyrosinases. Therefore, most tyrosinase inhibitors of melanogenesis have been designed based mushroom tyrosinase due to its low price and business availability [13,14]. Since tyrosinases inhibit melanogenesis, they have been studied as cosmetic agents. Most commercially available skin-lightening products are based on tyrosinase inhibitors, and several promising tyrosinase inhibitors have been used for pharmaceutical, cosmeceutical, or agricultural purposes [15,16]. However, only a handful of compounds have been used clinically because of inadequate efficacy or some undesirable effects, such as possible carcinogenicity. Arbutin—the natural derivative of hydroquinone extracted from plants such as bearberries, blueberries, and cranberries, azelaic acid—a naturally occurring acid found in grains such as barley, wheat, and rye, hydroquinone contained in certain fruits, such as pears and blueberries, and kojic acid produced by several species of fungi are used as skin-lightening products for scientific and beauty applications [17,18,19]. However, those products have raised various health and safety concerns, which consist of renal [20], immune system, bone marrow [21], and melanocyte [22] toxicities, and possible carcinogenicity [23]. Therefore, novel tyrosinase inhibitors are needed that effectively cope with those undesirable effects. Additionally, in pharmaceutical and cosmeceutical applications, tyrosinase inhibitors are applied in food bioprocessing, and environmental technologies. Recently, tyrosinase protein has been used to produce tailor-made melanin and other polyphenolic materials to develop organic semiconductors and photovoltaic organic products, and to produce bioengineered biocatalysts for industry.

It has been observed that L-tyrosine and L-dihydroxyphenylalanine (L-DOPA) are recognized as substrates and intermediates in melanogenesis [24,25,26]. However, it has also be found that in multiple melanoma/melanocyte lines, L-tyrosine leads only to increase of melanin pigmentation without apparent changes in the expression of proteins/genes of the melanin synthesis pathway [24]. Other studies have shown that L-tyrosine and L-DOPA, in addition to being substrates and intermediates of melanogenesis, play bioregulatory roles as inducers and positive regulators of melanogenesis and other cellular processes [24,27]. The mechanistic signaling pathway of ligands has been depicted in Figure 1. Prior data reported that most of chemical compounds directly inhibit tyrosinase activity to cure melanin-associated disorders [19]. Tyrosinase controls the conversions of L-tyrosine to L-Dopa and L- Dopa to Dopa-quionone, and results in eumelanin and pheomelanin, respectively. The accumulation of eumelanin/pheomelanin beneath the skin causes melanin-associated diseases. Therefore, the inhibitory potential of compounds with tyrosinase may stop the downstream signaling pathway and may help to cure melanin-associated disorders [28,29].

The biological effects of hormones on human melanocytes are still being investigated [30]. There are multiple hormones that have mitogenic and/or melanogenic effects on human melanocytes [31]. For example, melanotropic hormones, such as α-melanocyte-stimulating hormone (α-MSH) and adrenocorticotrophic hormone (ACTH), are responsible for skin darkening. The mechanism of up-regulation of tyrosinase activity in human normal melanocytes by melanotropins through the cyclic AMP (cAMP) signaling pathway is now recognized to play a key role in the regulation of skin pigmentation. α-MSH is known to elevate intracellular cAMP levels through the melanocortin 1 receptor (MC1R) and this plays a critical role in the regulation of melanogenesis [30]. Two important female hormones estrogen and progesterone are also responsible for skin pigmentation during pregnancy due to their increased levels in female body [32]. Melasma commonly known as mask of pregnancy develops on the skin due to imbalance of these hormonal changes which ultimately results in dark patches on the body particularly on both sides of the face and pelvic regions [33]. Moreover, local changes in skin pigmentation such as age spots (solar lentigines) can occur with ageing [34]. 

The other skin disorders including acne, atopic dermatitis or lichen planus occur by localized post-inflammatory hyperpigmentation (PIH) resulting from overproduction of melanin or abnormal melanin deposition in the epidermis or dermis regions [35]. Furthermore, the patients suffering from Addison’s disease which is primary known as adrenal insufficiency also suffer from hyperpigmented skin due to elevated levels of Adrenocorticotropic hormone (ACTH), a polypeptide tropic hormone produced and secreted by the anterior pituitary gland that stimulates adrenal glands to release cortisol [36]. Vitiligo is an acquired pigmentary disorder of unknown etiology characterized by the localized loss of skin color [37]. Moreover, several genetic diseases, called genodermatoses, affect skin pigmentation. One example is albinism, which includes a group of inherited disorders that are characterized by little or no production of the pigment melanin, impacting skin, hair and eye color [38]. Melatonin is another hormone produced in the *glandula pinealis* that follows a circadian light-dependent rhythm of secretion. Melatonin is implicated in skin functions such as hair cycling and fur pigmentation, and melatonin receptors are expressed in many skin cell types including normal and malignant keratinocytes, melanocytes and fibroblasts [39,40]. 

The basic aim of this article is to review various novel tyrosinase inhibitors which have been extracted from plants or synthesized in chemical laboratory to address melanogenesis. Additionally, another significant aspect is the development of skin whitening agents which is a basic pillar of cosmetic industry. The present article reviews chemical compounds along with their structures and inhibitory potentials against tyrosinase and is intended for the medicinal chemists to screen out and to synthesize novel chemical inhibitors. 

## 2. Tyrosinase Inhibitors against Melanogenesis

### 2.1. Simple Phenolic Derivatives

Phenolic compounds having at least one fragrant ring and one (or more) hydroxyl organization may be categorized primarily based on the quantity of their carbon atoms and association among them [29]. There is a large variety of phenolic compounds from small to large and complicated tannins and derived polyphenols differing by their molecular weight and quantity of fragrant-rings and mostly all of these products are promising inhibitors of melanogenesis [41]. There are multiple phenolic derivatives which have been reported in different studies in simple or conjugated forms [42,43,44,45,46]. Chen et al. observed that alkylhydroquinone 10’(Z)-heptadecenylhydroquinone, obtained from the sap of the lacquer tree *Rhus succedanea*, can inhibit the activity of tyrosinase and suppress melanin production in animal cells. The half-maximal inhibitory concentration IC_50_ of this compound (37 µM) is much less than the IC_50_ of hydroquinone (70 µM), which is an acknowledged inhibitor of tyrosinase (Figure 2). They have concluded that the potent inhibitory impact of this compound on tyrosinase activity is possibly due to its heptadecenyl chain, which allows the oxidation of the hydroquinone ring [43,44].

Isotachioside, a chemical compound used to treat melanogenesis found in plants *Isotachis japonica* and *Protea neriifolia*, and its glycoside derivatives are categorized as analogs of arbutin, a standard molecule used for tyrosinase inhibition screening. The isotachioside derivatives such as glucoside, xyloside, cellobioside and maltoside are acting as tyrosinase inhibitors [47,48]. Among those novel inhibitors, glucoside derivative (IC_50_  =  417 µM) is the most effective, indicating that the structural mixture of resorcinol and glucose induces enormous inhibitory effect [29,49]. Hydroquinone derivatives, inclusive of α and β-arbutin, are recognized as tyrosinase inhibitors [50,51]. Earlier studies showed that deoxyarbutin, along with its derivatives, can be used to ameliorate hyperpigmented lesions or lighten pores and skin due to much less toxicity and powerful inhibitory effects [52,53]. Plants produce a large diverse class of polyphenols including phenolic acids, flavonoids, stilbenes and lignans which can be used as weak or potent inhibitors of tyrosinase [54].

### 2.2. Flavonoids

The flavonoid derivatives that are mainly present in natural plants are used as the best tyrosinase inhibitors [55,56,57]. There is a substantial correlation between the inhibitory efficiency of flavonoids on mushroom tyrosinase and melanin synthesis in melanocytes [58]. In searching for powerful tyrosinase inhibitors from herbal products, many flavonoid compounds have been extracted and evaluated for their inhibitory effects on mushroom tyrosinase [59]. Flavonoids are generally categorized into flavones, flavonols, isoflavones, flavanones, flavanoles and anthocyanidins, dihydroflavones, flavan-3,4-diols, coumarins, chalcones, dihydrochalcones and aurones [60]. Additionally, prenylated and vinylated flavonoids, such as flavonoid glycosides, represent different subclasses of flavonoids. Some flavonoid glycosides including myricetin 3-galactoside and quercetin 3-O-β-galactopyronaside, are used as supplements possessing antioxidant activity. Interestingly, it has been proven that deglycosylation of a few flavonoid glycosides by using far-infrared irradiation can increase their tyrosinase inhibitory activity [61]. 

#### 2.2.1. Flavones

The most common flavones are luteolin, apigenin, baicalein, chrysin and their glycosides [62]. It has been shown that apigenin and nobiletin from the methanolic extract of the heartwood of *Artocapus altilis* with 11 other phenolic compounds demonstrate inhibitory activities on tyrosinase [63]. In another work, it has been observed that derivative of flavone, called 7,8,4’-trihydroxyflavone inhibits diphenolase activity of tyrosinase with an IC_50_ value 10.31  ±  0.41 µM in a noncompetitive manner with the inhibitor constant *K_i_* of 9.50  ±  0.40 µM [19]. Computational analysis reported that the binding process involves hydrogen bonds and hydrophobic interactions between the ligands and the residues His244 and Met280 of active site [64]. Moreover, there are some different studies which also explore the similar binding results of different compounds at same binding position in the tyrosinase [53,65]. 

#### 2.2.2. Flavonoles

The major derivatives of flavonoles, such as myricetin, kaempferol, quercetin, morin, isorhamnetin, galangin and their glycosides, have been identified as tyrosinase inhibitors [66,67]. Kinetics studies show that morin reversibly inhibits tyrosinase through a multi-phase kinetic process and binds to tyrosinase at a single binding site mainly by hydrogen bonds and van der Waals interactions [68,69]. Furthermore, galangin, kaempferol and quercetin inhibit the oxidation of L-DOPA catalyzed by mushroom tyrosinase and presumably this inhibitory activity originates from their copper chelating ability [69]. Interestingly, quercetin behaves as a cofactor and does not inhibit monophenolase activity, while galangin inhibits monophenolase activity and does not act as a cofactor, whereas kaempferol neither acts as a cofactor nor inhibits monophenolase activity [70].

#### 2.2.3. Isoflavones

Isoflavones including daidzein and genistein, both occurring in soybeans, glycitein found in soy food products, formononetin occurring in green beans and clover sprouts, and their glycosides, are mostly present in medicinal herbs [71]. It has been shown that the natural derivatives of o-dihydroxyisoflavone are potent antioxidatives [72]. Additionally, glabridin found in the root extract of licorice (*Glycyrrhiza glabra*) with an IC_50_ of 0.43 µM exhibits excellent inhibitory effects on tyrosinase [73].

#### 2.2.4. Flavanones

Flavanone derivatives such as naringenin, hesperetin, eriodictyol and their glycosides and flavanonols (taxifolin) are mainly found in citrus fruits and the medicinal herbs [74]. A copper chelator flavanone named hesperetin, the main flavonoid in lemons and sweet oranges, inhibits tyrosinase activity reversibly and competitively [75]. The 8-anilino-1-naphthalenesulfonic acid (ANS)-binding fluorescence analysis shows that hesperetin disrupts tyrosinase structure by hydrophobic interactions. In addition, hesperetin chelates a copper ion coordinated with three histidine residues (His-61, His-85, and His-259) within the active site pocket of the enzyme, as shown by docking simulation studies [76,77,78]. 

## 3. Anthocyanidins and Curcuminoids

The most common anthocyanidins are cyanidin, delphinidin, malvidin, peonidin, petunidin, and pelargonidin, occurring in the medicinal herbs [79]. It has been observed that there is a significant correlation between anthocyanin content and inhibitory activity for both human and mushroom tyrosinases [80]. Furthermore, a couple of phenolic ligands such as curcumin and desmethoxycurcumin significantly inhibit activity of tyrosinase in comparison to standard kojic acid [81,82].

### 3.1. Coumarins

Coumarins found in many plants, especially in cassia cinnamon and tonka beans, are also known as good therapeutic inhibitors of tyrosinase [83]. The inhibitory effects of several coumarin derivatives including 3-aryl and 3-heteroarylcoumarins, esculetin, coumarinolignoid 8’-epi-cleomiscosin, umbelliferone, phenyl coumarins, hydroxycoumarins, thiophosphonic acid diamides, diazaphosphinanes coumarin derivatives, cardol-coumarin derivatives and coumarin-resveratrol hybrids, have been reported in the literature [84,85,86]. 

### 3.2. Chalcones and Dihydrochalcones

Chalcones (butein, phloretin, sappan-chalcone, carthamin, etc.), or 1,3-diphenyl-2-propen-1-ones, are some of the most important classes of flavonoids occurring in foods, vegetables, tea, and spices [54]. It has been shown that some natural and synthetic chalcones and their derivatives are identified as new potent depigmentation agents and tyrosinase inhibitors. Synthetic chalcones and their derivatives evaluated by various research studies include oxindole-based chalcones [87], 1-(2-cyclohexylmethoxy-6-hydroxy-phenyl)-3-(4-hydroxymethyl-phenyl) propenone derivatives [88] and isoxazole chalcone derivatives [89]. It has been observed that the efficacy of a chalcone depends upon the location of the hydroxyl groups on both aromatic rings as well as the presence of a catechol moiety, and it does not correlate with increasing tyrosinase inhibition potency [90].

### 3.3. Stilbenes

The best-known stilbene that inhibits tyrosinase activity is resveratrol occurring in grapes and red wine [54]. There are multiple stilbene derivatives from natural and synthetic sources that have been investigated for their tyrosinase inhibition activity [91,92]. However, enzymatic assay studies have shown that resveratrol does not inhibit the diphenolase activity of tyrosinase, but L-tyrosine oxidation by tyrosinase was suppressed in presence of 100 µM resveratrol [93,94]. Oxyresveratrol a stilbenoid found in the heartwood of *Artocarpus lakoocha* is also known as not an inhibitor but an alternative tyrosinase substrate [95].

## 4. Quinone and Phenyl Derivatives

Quinone and phenyl derivatives are small molecules that are mostly derived from aromatic compounds such as benzene or naphthalene. Among these compounds, aloin occurring naturally in various *Aloe* species, anthraquinones found in inter alia, in aloe latex, senna, rhubarb, and tanshinone IIA a natural product found in *Salvia miltiorrhiza* have been verified as tyrosinase inhibitors [96,97]. Additionally, multiple biphenyl derivatives such as 4,4’-dihydroxybiphenyl, biphenyl ester derivatives, biphenyl construction from flavan-3-ol substrates, hydroxylated biphenyls, functionalized bis-biphenyl substituted thiazolidinones, phenylbenzoic acid derivatives, phenylethylamide and phenylmethylamide derivatives, hydroxy substituted 2-phenyl-naphthalenes, 4-hydroxyphenyl beta-d-oligoxylosides, benzenethiol or phenylthiol, 2-((1Z)-(2–(2,4-dinitrophenyl)ydrazine-1-ylidene)methyl) phenol and 4-[(4-hydroxyphenyl)azo]-benzenesulfonamide, have been identified as tyrosinase inhibitors [98].

## 5. Pyridine, Piperidine, Pyridinones, Hydroxypyridinone, Azole and Thiazolidine Derivatives

Hydroxypyridinone derivatives, 3-hydroxypyridine-4-one derivatives hydroxypyridinone-L-phenylalanine and pyridinones have been characterized for their anti-tyrosinase activity [99]. Among these inhibitors, one mixed-type inhibitor from hydroxypyridinone-l-phenylalanine conjugates, named ((S)-(5-(benzyloxy)-1-octyl-4-oxo-1,4-dihydropyridin-2-yl) methyl 2-amino-3-phenylpropanoate), showed potent inhibitory effect with IC_50_ values of 12.6 and 4.0 µM for monophenolase and diphenolase activities, respectively [100]. There are various azole derivatives designed to inhibit tyrosinase activity [101,102]. The newly discovered types of inhibitors include DL-3(5-benzazolyl) alanines and alpha-methyldopa analogs, aryl pyrazoles, heterocyclic hybrids based on thiazolidinone scaffolds, pyrazolo [4,3-e] [1,2,4] triazine sulfonamides and sildenafil, 1,3-oxazine-tetrazole, indole-spliced thiadiazole, benzimidazole-1,2,3-triazole hybrids, 1,2,3-triazole-linked coumarinopyrazole conjugates [103,104,105]. Hassan et al. designed various triazole-based derivatives against tyrosinase (Figure 1) and their biological activity is shown in Table 1 [104].

## 6. Kojic Acid Analogs and Carboxylic Acids Derivatives

Kojic acid is a well-known tyrosinase inhibitor [106,107,108,109,110]. Some kojic acid derivatives despite their depigmenting properties did not display tyrosinase inhibitory activity. The prior study showed that a kojic acid analog 5-phenyl-3-[5-hydroxy-4-pyrone-2-yl-methylmercapto]-4-(2,4-dihydroxylbenzylamino)-1,2,4 triazol is a potent competitive tyrosinase inhibitor with an IC_50_ value of 1.35 ± 2.15 µM [54,111]. Inhibitory effects of pyruvic acid, acrylic acid, propanoic acid, 2-oxo-butanoic acid, and 2-oxo-octanoic acid, (S)- and (R)-6-hydroxy-2,5,7,8-tetramethylchroman-2-carboxylic acids have been investigated to check the tyrosinase inhibition [19,112,113]. The prior report showed that carvacrol derivatives also possessed good inhibitory potential against tyrosinase in comparison with standard Kojic acid (16.69 ± 2.8) (Table 2) [106].

## 7. Conclusions and Future Prospects

Tyrosinase is a vital enzyme involved in the browning of food and depigmentation disorders in humans [65]. Despite the diversity of naturally occurring and semi-synthetic and synthetic tyrosinase inhibitors that have recently been studied, there is still an urgent need to find compounds that are both highly effective and safe, without any noticeable side effects that could be widely applied in medicine, food industry and cosmetology. To achieve this goal, researchers have used appropriately designed scaffolds inspired by the structures of natural compounds and are developing novel synthetic inhibitors. In this review, different tyrosinase inhibitors have been discussed that exhibit promising therapeutic potential for treatment of melanogenesis. However, despite the existence of a wide range of tyrosinase inhibitors from both natural and synthetic sources, only a few of them, in addition to being effective, are known as safe, non-toxic compounds and probably with reduced side effects [54]. Therefore, screening such products in in vitro, in vivo and computational studies are the most important tasks to select potent inhibitors for treating melanogenesis. Taken together, the information provided in this review is the result of numerous efforts of many research groups studying this problem, which could lead in near future to the development of effective anti-tyrosinase agents with increased efficiency and safety in the food, pharmaceutical and cosmetics industries.

## Data Availability

Not applicable.

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
