# Peer review of "Tyrosinase Inhibitors Naturally Present in Plants and Synthetic Modifications of These Natural Products as Anti-Melanogenic Agents: A Review"

_molecules, 2023, doi:10.3390/molecules28010378_

Round 1

Reviewer 1 Report

This review summarizes the current research progress of tyrosinase inhibitor. The characteristic color of bacteria, fungi, plants and animals is very much related to melanin. This review provides a clear and detailed introduction to melanogenesis and melanin inhibitors, as well as a description of the possible side effects of some tyrosinase inhibitors. This review is of great significance for the subsequent development of safe tyrosinase inhibitors in the food industry, medicine industry, beauty industry and other daily necessities that people need. So I quite approve of this review, only some detailed modification suggestions are as follows:

1.      Is there an author missing at the end of the list?

2.      Line 12: There is no mention of drug design in the text, and the first sentence of the abstract is unfitted.

3.      Line 24: Some words in the title should not be also used as keywords, keywords should be other important words, such as “tyrosinase inhibitors, natural products, anti-melanogenic agents” change to “synthetic inhibitors, chemical compounds, anti-tyrosinase agents”

4.      Line 27: The first paragraph cites too few references, each sencence need a reference to support this idea.

5.      Line 63: Title 1.1 can be deleted, and only three paragraphs in the background so do not need to be subdivided.

6.      Line 97: “glucoside derivative (IC50 = 417 μM) is the most effective”. The smaller vaule of IC50, the stronger the inhibition effect, the vaule (IC50 = 417 μM) is larger than that mentioned in Line 87, please confirm the accuracy of the description.

7.      Line 109: The title structure of the whole article needs to be adjusted. For example, 2.1.2 Flavonoids: flavonoids are generally categorized into flavones, flavonols, isoflavones…….So 2.2 flavones should be the sub level of flavonoids.

8.      Line 162: Any tyrosinase inhibitors mentioned in this review should point out the IC50, this could give other readers or researchers a better idea of which one is more useful.

9.      Line 187 is missing the heading number.

10.  Line 195-229: These parts still belong to different types of tyrosinase inhibitors, that is the second part, “2. Tyrosinase inhibitors against melanogenesis” can be divided into different points depending on source, function, or strengths and weaknesses.

11.  Conclusion and Future prospects: The author mentioned some tyrosinase inhibitors show no side effects, these category inhibitors can be summarized in the article, which can give advice to other academics or developers.

References:

12.  The format of the references should be consistent with the format requirements of the journal, and all journal abbreviations should be italicized. Please check them all carefully.

Author Response

Reviewer 1. Comments and Suggestions for Authors

This review summarizes the current research progress of tyrosinase inhibitor. The characteristic color of bacteria, fungi, plants and animals is very much related to melanin. This review provides a clear and detailed introduction to melanogenesis and melanin inhibitors, as well as a description of the possible side effects of some tyrosinase inhibitors. This review is of great significance for the subsequent development of safe tyrosinase inhibitors in the food industry, medicine industry, beauty industry and other daily necessities that people need. So I quite approve of this review, only some detailed modification suggestions are as follows:

Answer: Thank you very much for your appreciation of our work and for suggestions how to improve the manuscript.  

  1. Is there an author missing at the end of the list?

Answer: All authors have been listed in the manuscript

  1. Line 12: There is no mention of drug design in the text, and the first sentence of the abstract is unfitted.

Answer: The first line of the abstract has been modified.

  1. Line 24: Some words in the title should not be also used as keywords, keywords should be other important words, such as “tyrosinase inhibitors, natural products, anti-melanogenic agents” change to “synthetic inhibitors, chemical compounds, anti-tyrosinase agents”

Answer: The keywords have been modified in the revised version of the manuscript.

  1. Line 27: The first paragraph cites too few references, each sentence need a reference to support this idea.

Answer: The new references have been incorporated in the manuscript, as suggested by the reviewer.

  1. Line 63: Title 1.1 can be deleted, and only three paragraphs in the background so do not need to be subdivided.

Answer: The title 1.1 has been deleted from the manuscript.

  1. Line 97: “glucoside derivative (IC50= 417 μM) is the most effective”. The smaller value of IC50, the stronger the inhibition effect, the value (IC50 = 417 μM) is larger than that mentioned in Line 87, please confirm the accuracy of the description.

Answer: Yes, we agree with the reviewer. Both compounds belong to different classes, one belongs to hydroquinone derivatives and the other to glucosidase derivatives.

  1. Line 109: The title structure of the whole article needs to be adjusted. For example, 2.1.2 Flavonoids: flavonoids are generally categorized into flavones, flavonols, isoflavones…….So 2.2 flavones should be the sub level of flavonoids.

Answer: The title structure has been modified in the revised manuscript.

  1. Line 162: Any tyrosinase inhibitors mentioned in this review should point out the IC50, this could give other readers or researchers a better idea of which one is more useful.

Answer: The glabridin from root extract of licorice (Glycyrrhiza glabra) is showing a good inhibitory potential on tyrosinase with IC50 = 0.43 µM. However, 5-phenyl-3-[5-hydroxy-4-pyrone-2-yl-methylmercapto]-4-(2,4-dihydroxylbenzylamino)-1,2,4 triazol is a potent competitive synthetic tyrosinase inhibitor with an IC50 value of 1.35 ± 2.15 µM.

  1. Line 187 is missing the heading number.

Answer: The heading number has been added in the manuscript.

  1. Line 195-229: These parts still belong to different types of tyrosinase inhibitors, that is the second part, “2. Tyrosinase inhibitors against melanogenesis” can be divided into different points depending on source, function, or strengths and weaknesses.

Answer: Thank you very much for a careful review of our manuscript. This section covers a different class of compounds; therefore, we have separated these derivatives in the manuscript.

  1. Conclusion and Future prospects: The author mentioned some tyrosinase inhibitors show no side effects, these category inhibitors can be summarized in the article, which can give advice to other academics or developers.

Answer: We have modified this section to explore different classes of tyrosinase inhibitors, and added references.

  1. The format of the references should be consistent with the format requirements of the journal, and all journal abbreviations should be italicized. Please check them all carefully.

Answer: The format of references has been modified in the revised manuscript, according to the journal requirements.

Reviewer 2 Report

Dear authors.

Plants attract the attention of researchers as a source of various useful substances with biological activity. Such studies are relevant and the results are interesting. The scientific content of the manuscript justifies its publication, but some additions and modifications will significantly improve the quality of the article.

Major comments:

1) In Introduction, the goal must be formulated.

2) What are the main areas of application of melanogenesis inhibitors?

3) Sections 4-8 should be supplemented or merged. A section of several lines has no right to exist.

4) The authors' critical view of the given information is lacking.

5) Figures and tables should improve the perception of information by readers. Figure and scheme are clearly not enough.

6) In the References, 23% of publications refer to 2017-2022 (the last 5 years); the remaining 77% of used sources are older than 5 years. It is recommended to increase the share of references to sources published over the last 5 years when analyzing the current state of research in the area under consideration, since this area of knowledge is rapidly developing.

Author Response

Reviewer 2

Dear authors.

Plants attract the attention of researchers as a source of various useful substances with biological activity. Such studies are relevant and the results are interesting. The scientific content of the manuscript justifies its publication, but some additions and modifications will significantly improve the quality of the article.

Major comments:

1) In Introduction, the goal must be formulated.

Answer: A new paragraph formulating our goals has been added in revised Introduction.

2) What are the main areas of application of melanogenesis inhibitors?

Answer: The main applications of melanogenesis inhibitors have been shortly reviewed in the revised manuscript.

3) Sections 4-8 should be supplemented or merged. A section of several lines has no right to exist.

Answer: A couple of sections have been merged in the revised manuscript.

4) The authors' critical view of the given information is lacking.

Answer: Two new tables have been added in the revised manuscript, that provide more objective information on inhibitory activities of various compounds.

5) Figures and tables should improve the perception of information by readers. Figure and scheme are clearly not enough.

Answer: We have incorporated a new figure and two tables in the revised manuscript to improve the perception of information by readers.

6) In the References, 23% of publications refer to 2017-2022 (the last 5 years); the remaining 77% of used sources are older than 5 years. It is recommended to increase the share of references to sources published over the last 5 years when analyzing the current state of research in the area under consideration, since this area of knowledge is rapidly developing.

Answer: New recently published references have been added into the manuscript to present the current state-of-the-art review of the field.

Round 2

Reviewer 2 Report

Dear Authors

My comments are taken into account
